# Mitotic H3K9ac is controlled by phase-specific activity of HDAC2, HDAC3, and SIRT1

Shashi Gandhi[1], Raizy Mitterhoff[1], Rachel Rapoport[1], Marganit Farago[1], Avraham Greenberg[1], Lauren Hodge[2], Sharon Eden[1], Christopher Benner[2], Alon Goren[2], Itamar Simon[1]

**Histone acetylation levels are reduced during mitosis. To study the mitotic regulation of H3K9ac, we used an array of inhibitors targeting specific histone deacetylases. We evaluated the involvement of the targeted enzymes in regulating H3K9ac during all mitotic stages by immunofluorescence and immunoblots. We identified HDAC2, HDAC3, and SIRT1 as modulators of H3K9ac mitotic levels. HDAC2 inhibition increased H3K9ac levels in prophase, whereas HDAC3 or SIRT1 inhibition increased H3K9ac levels in metaphase. Next, we performed ChIP-seq on mitotic-arrested cells following targeted inhibition of these histone deacetylases. We found that both HDAC2 and HDAC3 have a similar impact on H3K9ac, and inhibiting either of these two HDACs substantially increases the levels of this histone acetylation in promoters, enhancers, and insulators. Altogether, our results support a model in which H3K9 deacetylation is a stepwise process—at prophase, HDAC2 modulates most transcription-associated H3K9ac-marked loci, and at metaphase, HDAC3 maintains the reduced acetylation, whereas SIRT1 potentially regulates H3K9ac by impacting HAT activity.**

## Introduction

The eukaryotic cell cycle is composed of many cellular events. Throughout interphase, the genome and other cellular components are duplicated, and during mitosis, the cell is divided into two daughter cells. The relatively short phase of mitosis is tightly regulated as it is critical to ensure that the cellular identity and function are correctly relayed to daughter cells (McIntosh & Hays, 2016; Palozola et al, 2019). The major molecular changes occurring during mitosis include a striking decrease in transcription (Prescott & Bender, 1962), condensation of the chromatin (Antonin & Neumann, 2016), nuclear envelope breakdown (Robbins & Gonatas, 1964), and loss of long-range intrachromosomal interactions (Naumova et al, 2013; Dileep et al, 2015).

We have conducted a comprehensive study to assess the changes in histone modifications during mitosis (Javasky et al, 2018). Our observations, together with other studies (Van Hooser et al, 1998; Kruhlak et al, 2001; McManus & Hendzel, 2006; Kelly et al, 2010; Hsiung et al, 2015; Liang et al, 2015; Liu et al, 2017; Ginno et al, 2018; Behera et al, 2019; Kang et al, 2020; Pelham-Webb et al, 2021), support the following notions: (i) the global epigenetic landscape is preserved during mitosis; (ii) the mitosis phase encompasses global reduction in histone acetylation; and (iii) there is a dramatic increase in histone phosphorylation.

What causes histone deacetylation during mitosis? Histone acetylation is carried out by histone acetyl transferases (HATs) (Lee & Workman, 2007) and is removed by HDACs (Haberland et al, 2009). HDACs encompass a diverse set of deacetylases that are involved in the regulation of the acetylation levels of histones and many other proteins. There are 18 HDACs in mammals that are classified into four major classes based on their homology to yeast HDACs. This categorization includes HDAC1-3 and HDAC8 in class I, HDAC4-10 in class II, the sirtuins SIRT1-7 in class III, and HDAC11 in class IV (Seto & Yoshida, 2014). Although some evidence has linked specific HDACs to mitosis, the exact HDAC that conducts the mitotic deacetylations is not well characterized. HDAC3 was suggested to play a key role in human mitosis because knocking it down in human HeLa cells affects mitosis histone deacetylation (Li et al, 2006). In yeast, the Hst2p histone deacetylase (SIR2 homolog), was shown to be responsible for the deacetylation of H4K16 during mitosis (Wilkins et al, 2014). In addition, the mitosis-specific deacetylation of histones could be carried out by a change in the balance between HATs and HDACs. Evidence supporting this notion was provided by a mass spectrometry–based study that determined the changes in DNA-associated proteins between various cell cycle stages and demonstrated a general retention of HDACs and depletion of HATs in mitosis (Ginno et al, 2018).

Here, we employed an array of small-molecule inhibitors, each one targeting a different set of HDACs or a specific HDAC (Table 1). We then measured the impact of these perturbations on deacetylation of H3K9 by immunofluorescence, immunoblots, and ChIP-seq. We observed that three histone deacetylases, namely—HDAC2,

---

[1]Department of Microbiology and Molecular Genetics, Institute of Medical Research Israel-Canada, Faculty of Medicine, The Hebrew University, Jerusalem, Israel
[2]Department of Medicine, University of California, San Diego, La Jolla, CA, USA

Correspondence: itamar.simon1@mail.huji.ac.il

**Table 1.  HDACs inhibitors and their specificity.**

| Name | Specificity | Concentration used | References |
|---|---|---|---|
| TSA | Pan-HDACs | 150 nM | Khan et al (2008) and Lobera et al (2013) |
| CHIDAMIDE | HDAC1,2,3,10 | 10 µM | Ning et al (2012) |
| RGFP966 | HDAC3 | 40 µM | Malvaez et al (2013) |
| CAY10683 | HDAC2 and HDAC6 | 1 µM | Pavlik et al (2013) and Bhattad et al (2020) |
| MS-275 | HDAC1,2,3 | 1 µM | Khan et al (2008) |
| NICOTINAMIDE | Pan-sirtuins | 10 mM | Hu et al (2014) |
| EX-527 | SIRT1 | 5 nM | Hu et al (2014) |

HDAC3, and SIRT1—are involved in H3K9 deacetylation. Further dissection of the roles each histone deacetylase plays revealed that HDAC2 initially deacetylates H3K9 at prophase, whereas HDAC3 activity is only detected later during metaphase. Although we noticed that SIRT1 is absent from the mitotic chromatin, we observed that this histone deacetylase potentially affects H3K9 acetylation indirectly through the modulation of mitotic HAT activity. Taken together, our results provide insight into the biochemical pathways involved in histone deacetylation during mitosis and pave the way for future studies aimed at deciphering the role the deacetylation process plays in regulation of mitotic gene expression and open chromatin.

## Results

### Identification of the deacetylases involved in modulating H3K9ac during mitosis

To study the changes in histone acetylation during mitosis, we focused on H3K9ac, which is reduced approximately by 2.7-fold during mitosis in HeLa-S3 cells (Javasky et al, 2018). To better identify the mitotic stage in which deacetylation takes place, we enriched HeLa-S3 cells for mitotic cells by releasing the cells from a double thymidine block for 8.5 h. This resulted in primarily G2/M cells with ~35% of the cells in mitosis, most of them in the metaphase stage (Figs 1A and S1). Immunofluorescence of H3K9ac at different mitotic stages revealed that the H3K9ac mark declines at prophase, reaches a minimal level at metaphase, remains low at anaphase, and gradually increases at later stages of mitosis (Fig 1B).

To identify the HDACs involved in the decrease of H3K9ac levels in metaphase, we arrested the cells at a metaphase-like stage with monoastral spindles using the kinesin 5 inhibitor STC (Skoufias et al, 2006). We added various HDAC inhibitors for 9 h (Fig 1C and Table 1) and measured H3K9ac levels by immunofluorescence (Fig 1D). We found that treatment with either the pan-HDAC inhibitor trichostatin A (TSA) (Khan et al, 2008; Lobera et al, 2013) or the pan-sirtuin inhibitor nicotinamide (NAM) (Hu et al, 2014) induces a significant increase ($P < 10^{-23}$; one sided $t$ test) in H3K9ac levels.

We further studied the deacetylation process by using specific inhibitors for particular histone deacetylases. Building on the results of previous studies linking HDAC3 and SIRT1 to mitosis (Li et al,

2006; Fatoba & Okorokov, 2011), we decided to first investigate the involvement of these histone deacetylases. We observed that the HDAC3-specific inhibitor RGFP966 (Malvaez et al, 2013) and the SIRT1-specific inhibitor EX-527 (Hu et al, 2014) induce an increase in mitotic H3K9ac that is similar to the acetylation levels following inhibition with pan-HDAC or pan-sirtuin inhibitors (Fig 1D). Similar results were obtained by immunoblots (Fig 1E).

The deacetylation of H3K9 initiated at prophase (Fig 1B); consequently, it cannot be fully studied by STC synchronization that arrests cells at a metaphase-like stage. We therefore targeted double thymidine synchronized cells with the small-molecule inhibitors. The various inhibitors (Table 1) were used upon release from the second thymidine block, and the level of H3K9ac was assessed 8.5 h later, a time-point that is maximally enriched for cells from all mitotic stages and cytokinesis (Figs 1A and S1). Cells were binned according to the different mitosis stages (prophase, metaphase, anaphase, and telophase) and cytokinesis, and H3K9ac immunofluorescence intensity was measured separately for each stage (Figs 2A and S2). We found that the general HDAC inhibitor TSA (Khan et al, 2008; Lobera et al, 2013) affects H3K9ac at prophase, whereas the pan-sirtuin inhibitor NAM (Hu et al, 2014) induces an increase in acetylation levels only in metaphase.

To identify the specific enzyme or enzymes that are involved in mitotic deacetylation, we used small molecules that selectively target a single or a concrete subset of histone deacetylases (Table 1). In particular, we observed that treatment with RGFP966, an HDAC3-specific inhibitor (Malvaez et al, 2013), affects H3K9ac levels only in metaphase, whereas treatment with chidamide that selectively inhibits HDAC1, HDAC2, HDAC3, and HDAC10 (Ning et al, 2012) shows very similar results to TSA. These results suggest that the deacetylation we observed at prophase is probably carried out by HDAC1, HDAC2, or HDAC10, given the differential impact of inhibiting HDAC3 alone by RGFP966.

To determine which of the three HDACs (HDAC1, HDAC2, or HDAC10) is modulating H3K9ac during prophase, we used two specific inhibitors—MS-275 that specifically targets HDAC1, HDAC2, and HDAC3 (Tatamiya et al, 2004; Khan et al, 2008) and CAY10683, an HDAC2 inhibitor (Pavlik et al, 2013; Bhattad et al, 2020). Following the specific inhibition, we measured H3K9ac levels in cells captured at different stages of mitosis (Fig 2A). We observed that both MS-275 and CAY10683 induced an increase in H3K9 acetylation levels in prophase. However, only MS-275 induced an increase in H3K9 acetylation levels in metaphase as well. This specific pattern

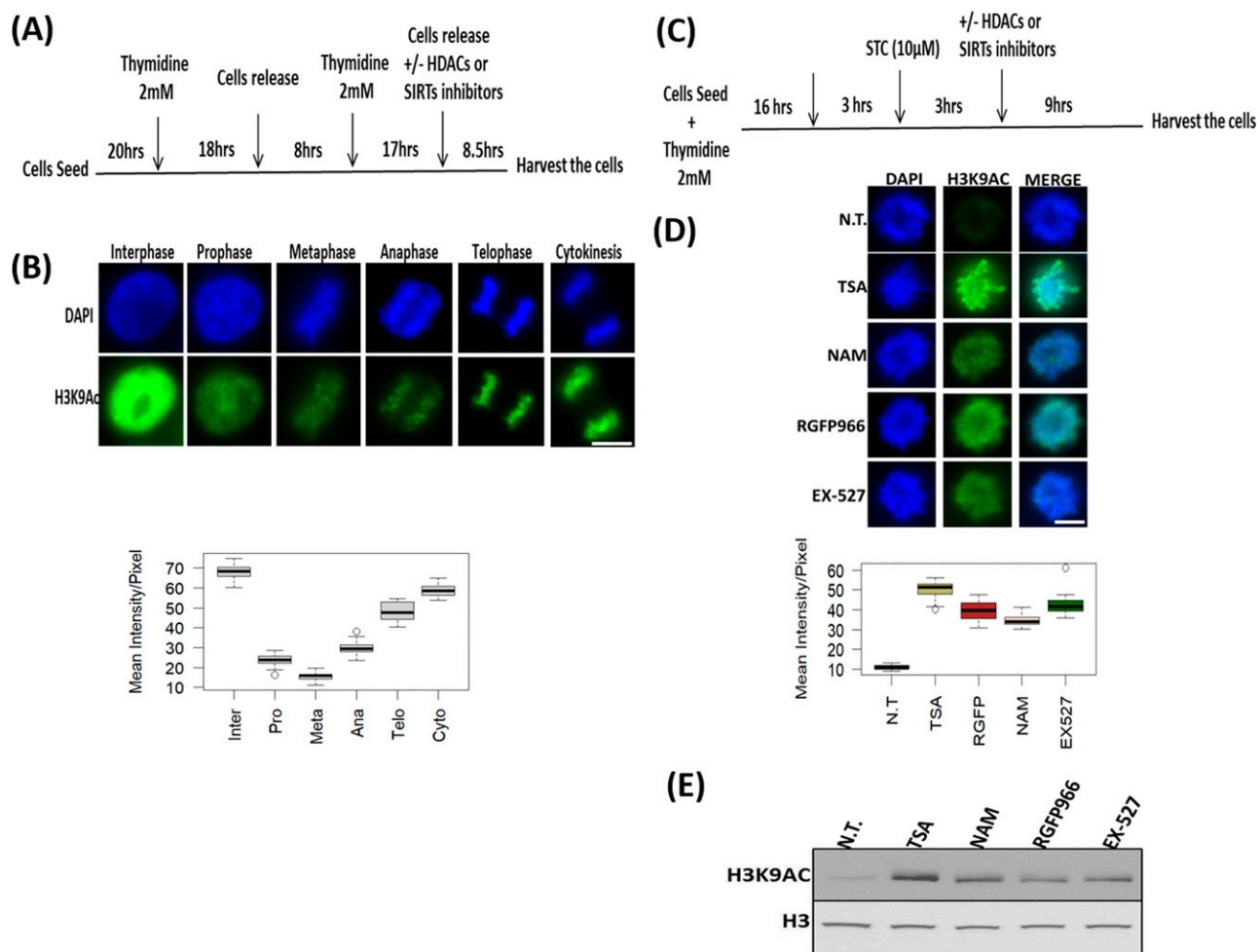

**Figure 1. H3K9ac levels during mitosis.**
**(A)** Schematic representation of the synchronization strategy by double thymidine block method. **(B)** Immunofluorescence of HeLa-S3 cells enriched for mitotic stages stained for H3K9ac (green) and DNA (blue). Representative pictures from the major mitosis stages are shown. Quantification of the immunofluorescence results presented below the pictures. Intensity values represent the mean ± SEM of at least 20 cells for every stage. **(C)** Schematic representation of the synchronization strategy by kinesin-5 inhibitor S-trityl-L-cysteine (STC). **(D)** Immunofluorescence of STC arrested HeLa-S3 cells stained for DNA (blue) and H3K9ac (green). The cells were either not treated (NT) or treated with TSA (150 nM); nicotinamide (NAM, 10 mM); RGFP966 (40 $\mu$M); EX-527 (5 nM). Quantification of the immunofluorescence results are shown below the pictures. Intensity values represent the mean ± SEM of at least 25 cells for every condition. **(D, E)** All treatments significantly ($P < 10^{-23}$; one sided $t$ test) increased H3K9ac levels (E) Immunoblot showing the H3K9ac signal for synchronized HeLa-S3 cells treated with the indicated inhibitors (same concentration as in D), total histone H3 serves as a loading control. Scale bars: 5 $\mu$m.

suggests that HDAC2 (targeted by CAY10683 and MS-275) deacetylates H3K9ac during prophase, whereas HDAC3 (targeted by RGFP966 and MS-275), which is not active in prophase, maintains the low acetylation levels from metaphase through telophase. Repeating the measurements with a larger sample of cells revealed that the CAY10683 inhibitor was able to induce a small increase in H3K9ac in metaphase as well (Fig S3).

As mentioned above, the pan-sirtuin inhibitor nicotinamide (NAM) induces a significant increase in H3K9ac levels as well. To study the involvement of the NAM-target sirtuins in H3K9 deacetylation in mitosis, we used the SIRT1-specific inhibitor EX-527 (Hu et al, 2014). The effect of SIRT1 inhibition was very similar to that of pan-sirtuin inhibition, suggesting that among the sirtuins, SIRT1 is the main mitotic H3K9 deacetylase.

We then evaluated whether the results we obtained in HeLa-S3 can be reproduced in another system by repeating the main experiments in MEFs. To this end, we synchronized the MEFs with a double thymidine block and measured H3K9ac levels by immunofluorescence 8.5 h after release from the second block. We focused on prophase and metaphase in the MEFs and observed high similarity to the results obtained in the human cell line (HeLa-S3). Specifically, treatment of MEFs with TSA, chidamide, or MS-257 induced an increase in H3K9ac levels in both prophase and metaphase (Fig 2B). On the other hand, the increased H3K9ac level in MEFs was seen only in metaphase when treated with NAM, EX-527, or RGFP966 and only in prophase when treated with CAY10683 (Fig 2B).

Taken together, these results suggest that mitosis-associated H3K9 deacetylation is a two-stage process—HDAC2 performs

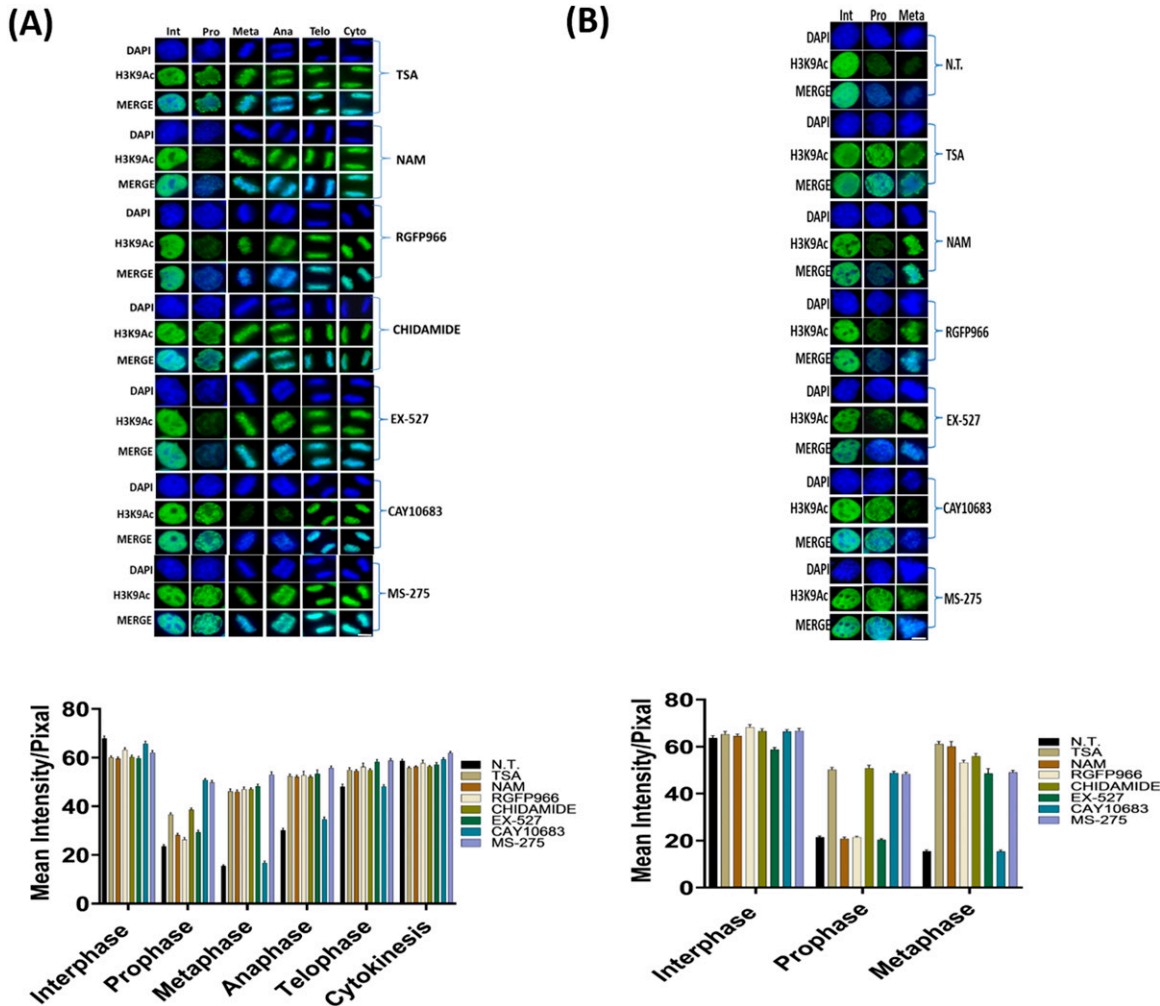

**Figure 2. HDAC inhibition shows involvement of specific HDACs in modulating the dynamics of H3K9ac during mitosis.**
**(A)** Immunofluorescence of HeLa-S3 cells enriched for mitotic stages stained for H3K9ac (green) and DNA (blue). Representative pictures from the major mitosis stages are shown. The cells were either not treated (NT) or treated with TSA (150 nM); nicotinamide (NAM, 10 mM); RGFP966 (40 $\mu$M); chidamide (10 $\mu$M); EX-527 (5 nM); CAY10683 (1 $\mu$M); MS-275 (1 $\mu$M). Below: quantification of the immunofluorescence results. Intensity values represent the mean ± SEM of at least 20 cells for every stage.
**(B)** Immunofluorescence of MEFs cells enriched for mitotic stages stained for H3K9ac (green) and DNA (blue). Representative pictures from the interphase, prophase, and metaphase stages are shown. The cells were either not treated (NT) or treated with TSA (150 nM); nicotinamide (NAM, 10 mM); RGFP966 (40 $\mu$M); chidamide (10 $\mu$M); EX-527 (5 nM); CAY10683 (1 $\mu$M); MS-275 (1 $\mu$M). Below: quantification of the immunofluorescence results. Intensity values represent the mean ± SEM of at least 20 cells for every stage. All treatments (both in HeLa-S3 and MEF cells) beside CAY10863 significantly affect H3K9ac levels at metaphase ($P < 10^{-15}$, FDR corrected $t$ test). Scale bars: 5 $\mu$m. For a boxplot version of the graphs, see Fig S3.

deacetylation during prophase, and HDAC3 and SIRT1 subsequently maintain low acetylation levels in metaphase.

## Variations in the mitotic chromatin association of key histone deacetylases and histone acetyltransferases

Our finding that the deacetylation process is dependent on HDAC2, HDAC3, and SIRT1, suggests that these deacetylases are associated with the mitotic chromosomes. We carried out immunofluorescence experiments targeting key HDACs and sirtuins to evaluate their mitotic localization. Using our double thymidine synchronization scheme (Fig 1A), we observed that among the HDACs tested, mainly HDAC3 was retained on the mitotic chromosomes during all mitotic stages. On the other hand, HDAC1 and HDAC2 appeared to be associated with the

mitotic chromosomes at lower levels during metaphase (Fig 3A). Surprisingly, even though we observed that inhibition of SIRT1 brings about an increase in H3K9ac during metaphase, neither SIRT1 nor the other sirtuins we tested (SIRT2, SIRT6, and SIRT7) showed strong binding to the chromatin in metaphase. Although we observed low binding of SIRT1 during prophase, there was a reduction in its chromatin localization starting at metaphase for the duration of mitosis (Fig 3B).

We performed similar experiments with several HATs and observed that HAT1 and EP300, but not CBP and GCNL2, retained their binding to the mitotic chromosomes. Interestingly, both HAT1 and EP300 seem to be more spread out on the metaphase chromsomes than HDAC3 (Fig 3C). Next, we validated these results by immunoblotting of chromatin from metaphase (STC-arrested) and interphase cells. In a similar manner to the immunofluorescence experiments, we detected a

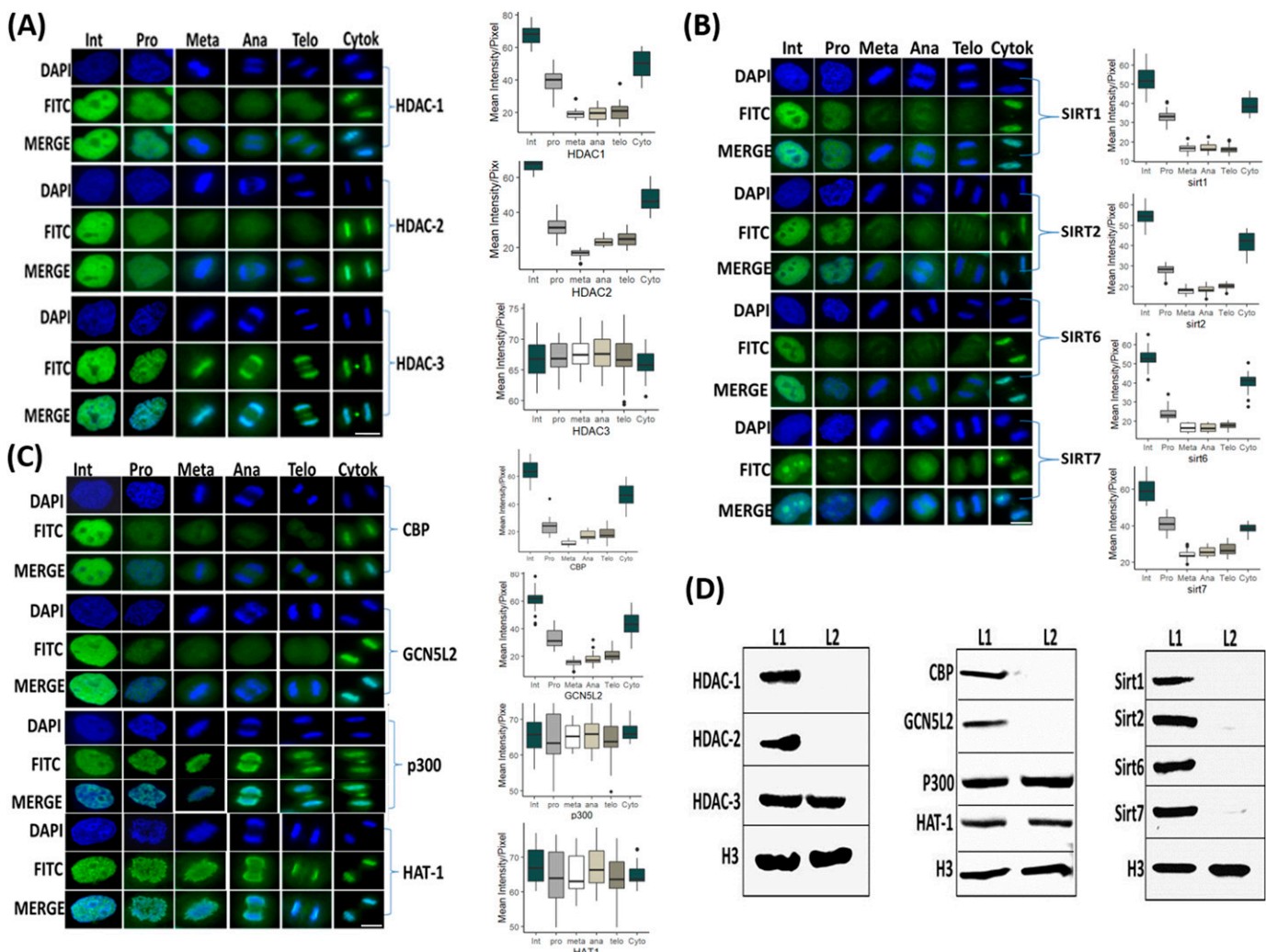

**Figure 3. Identification of the chromatin localization patterns of key histone acetyl transferases (HATs) and HDACs during mitosis.**
**(A, B, C)** Immunofluorescence of HeLa-S3 cells enriched for mitotic stages, stained for DNA (blue) and for various HDACs (A), sirtuins (B), and HATs (C) (green). Representative pictures from the major mitosis stages and interphase are shown, along with a quantification of at least 20 cells from each stage. Intensity values represent the mean ± SEM. Note that only HDAC3, P300, and HAT-1 are retained on the mitotic chromatin along all mitotic stages, whereas the other proteins show partial retention at prophase. **(D)** All factors (beside HDAC3, P300, and HAT1) show significant reduction at metaphase ($P < 10^{-16}$, FDR-corrected $t$ test) (D) Immunoblots showing the abundance of the indicated chromatin modifiers in the chromatin fraction of interphase (L1) and mitotic (L2) HeLa-S3 cells. Total histone H3 serves as loading control. Scale bars: 5 $\mu$m.
Source data are available for this figure.

signal only for HDAC3, HAT1, and EP300 from the metaphase chromatin fraction. On the other hand, using mitotic chromatin we could not identify a signal for HDAC1; HDAC2; SIRT1, 2, 6, and 7; or two of the HATs—CBP and GCNL2 (Fig 3D). Together, the localization of HDAC2 to mitotic chromatin at prophase, and the binding of HDAC3 throughout mitosis are in line with the two-stage model presented above, whereas the binding of SIRT1 at prophase was unexpected.

### SIRT1 potentially impacts H3K9ac levels indirectly by modulating HAT activity

We considered two hypotheses to reconcile our seemingly contradictory observations, namely, that SIRT1 inhibition impacts H3K9ac levels during mitosis while also showing low binding to chromatin

during mitosis. One option is that SIRT1 directly deacetylates H3K9 before metaphase, before it detaches from the mitotic chromosomes. However, we ruled out this possibility because inhibition of SIRT1 by EX-527 does not appear to impact the prophase levels of H3K9ac (Fig 2A). The second option we considered is that SIRT1 indirectly impacts the H3K9ac levels at metaphase by modulating the activity of other histone deacetylases or acetyl transferases.

We developed an in vitro assay to study the impact of SIRT1 on HAT activity in metaphase (Fig 4). We isolated protein extracts from STC-arrested HeLa-S3 cells either treated or untreated with the SIRT1 inhibitor EX-525. The extracts were used to acetylate His-tagged histone H3 peptide, and HAT activity was studied via two approaches. First, we used a colorimetric assay that measures the amount of Co-A released in the acetylation reaction via the DTNB

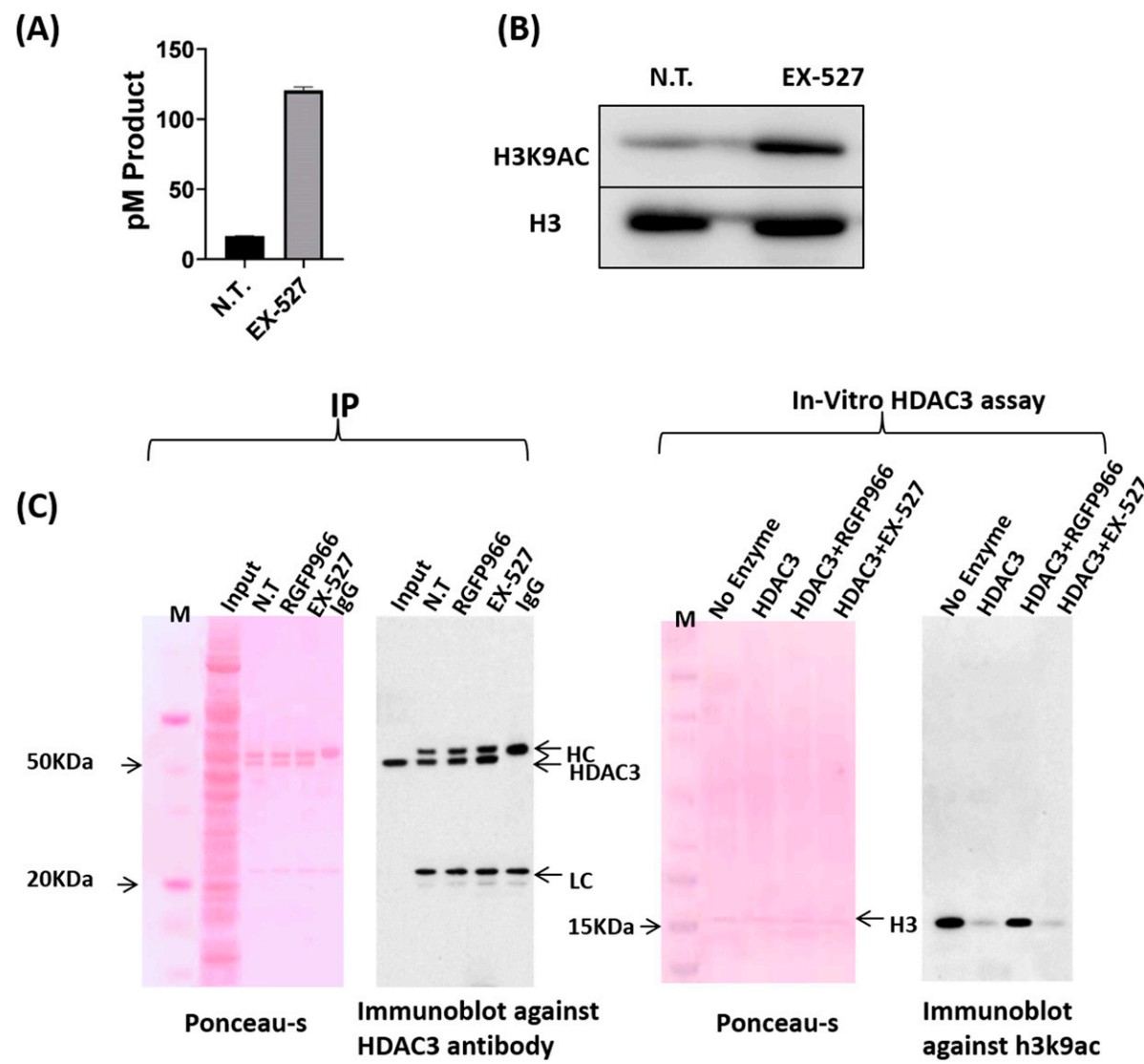

**Figure 4.  An in vitro assay supports an indirect modulation H3K9ac levels by impacting histone acetyl transferase (HAT) activity.**
**(A)** HAT activity colorimetric assay results (mean ± standard error, N = 2) for HeLa-S3 cells either not treated (NT) or treated with EX-527 (5 nM). **(B)** Immunoblot analysis showing H3K9 acetylation on His-tagged H3 after HAT activity of HeLa mitotic chromatin either not treated (NT) or treated with EX-527 (5 nM) on His-tagged H3. Total histone H3 serves as loading control. **(C)** Immunoprecipitation of HDAC3 tested by immunoblotting. HDAC3 antibodies and protein A magnetic beads were used to immunoprecipitate (IP) HDAC3 complexes from 1 mg of HeLa mitotic extract either not treated (NT) or treated with RGFP966 (40 $\mu$M) or EX-527 (5 nM). (i) Showing Ponceau S staining of HDAC3 immunocomplex; (ii) showing immunoblot using HDAC3 antibodies developed by ECL method; (iii) showing Ponceau S staining of HDAC3 activity assay reaction subjected to immunoblot; and (iv) showing immunoblot analysis of immunoprecipitated HDAC3 activity on acetylated his-tagged H3 protein.

absorbance at 412 nm (Foyn et al, 2017). In addition, we immuno-precipitated the His-tagged histone H3 peptide using nickel beads and evaluated the acetylation level of this H3 peptide by immu-noblot using an H3K9ac antibody. Both assays revealed that HAT activity was increased in extracts made from cells treated with the SIRT1 inhibitor Ex-527 (Fig 4A and B), suggesting that SIRT1 represses the cellular HAT activity during the metaphase stage.

Next, to test whether SIRT1 is impacting a histone deacetylase, we focused on HDAC3. Our reasoning was that HDAC3 was the only histone deacetylase we detected on the metaphase chromatin (Fig 3) and that HDAC3 is highly involved in regulating metaphase H3K9ac (Fig 2A). To evaluate the impact of SIRT1 on HDAC3 activity in metaphase, we immunoprecipitated HDAC3 from STC-arrested mitotic cells treated or untreated with EX-527. We then used acetylated His-tagged histone H3 peptide and measured H3K9ac levels by immunoblot. We observed that inhibition of SIRT1 does not impact the deacetylase activity of HDAC3 toward H3K9ac (Fig 4C). Taken together, we show that SIRT1 appears to indirectly affect H3K9ac levels by potentially modulating mitotic HAT activity. This result is in line with the observation that SIRT1 is mostly detached from the chromosomes during mitosis (Fig 3B).

## HDAC2 and HDAC3 activity regulate both common and unique genomic loci

The results described above suggest that deacetylation of H3K9 is conducted by HDAC2 in prophase and by HDAC3 in metaphase, with

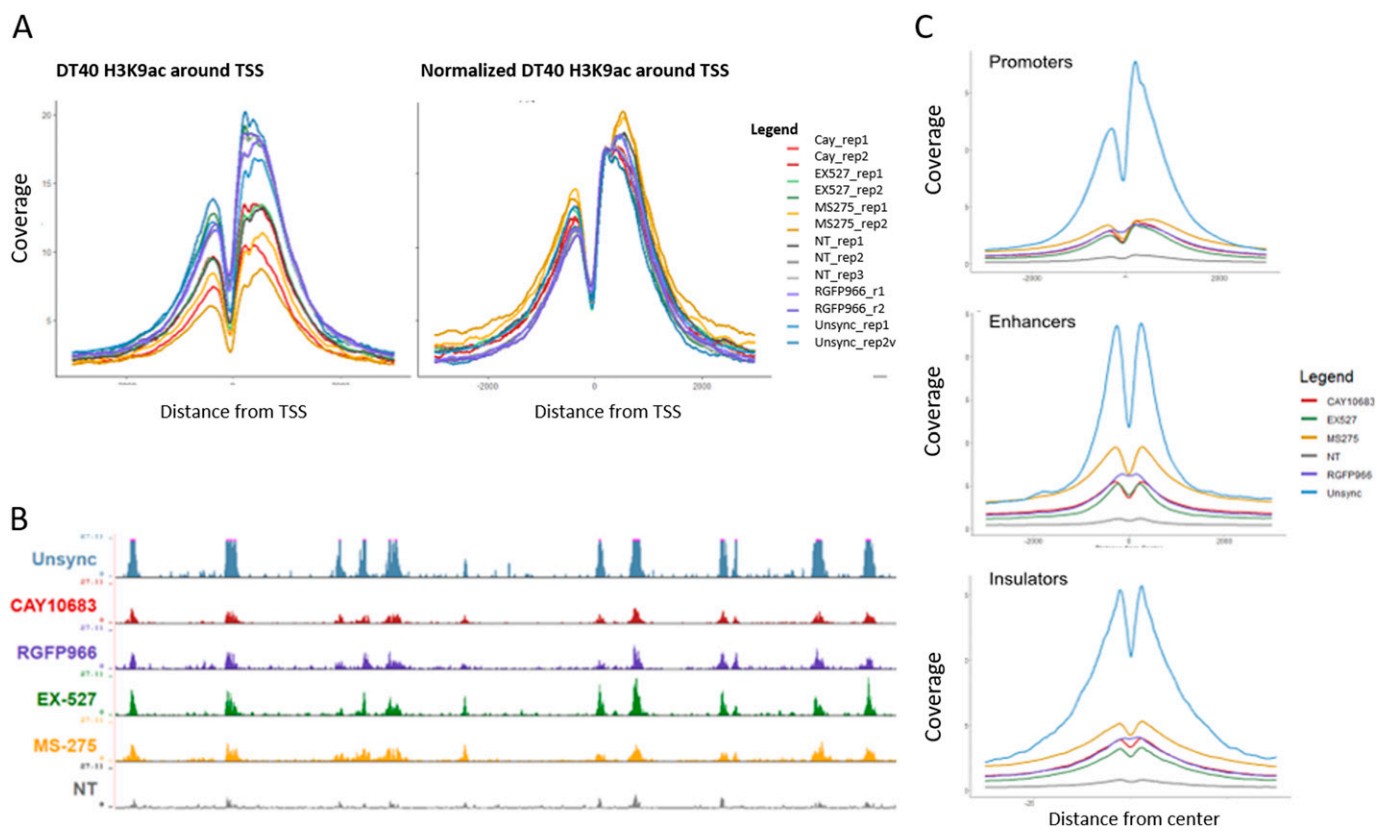

**Figure 5. Detection of genomic patterns associated with the activity of HDAC2 and HDAC3.**
**(A)** Metagene plots showing H3K9ac promoter occupancy in chicken DT-40 cells before (left) and after (right) normalization. **(B)** Genomic viewer (IGV) tracks representing the H3K9ac ChIP-seq enrichment for the indicated conditions. **(C)** Metagene plots showing H3K9ac occupancy around promoters, enhancers and insulators in HeLa-S3 cells. The data were normalized using the DT-40 promoter data. Similar results were obtained in a biological repeat (Fig S5).

SIRT1 indirectly involved in an additional reduction in H3K9ac during metaphase via modulation of HAT activity. However, these results were obtained by immunofluorescence and immunoblots and thus do not provide information about the regulation of histone acetylation at a genomic level by these deacetylases.

To determine genomic context of H3K9 deacetylation, we performed ChIP-seq for H3K9ac on STC-arrested cells, either treated or untreated with CAY10683 (HDAC2 inhibitor), RGFP966 (HDAC3 inhibitor), MS-275 (HDAC2 and HDAC3 inhibitor), and EX-527 (SIRT1 inhibitor) (Table 1). To quantitatively compare H3K9ac enrichment between the ChIP-seq experiments, we used the ChIP-Rx approach (Orlando et al, 2014) which normalizes the efficiency of ChIP by adding the same amount of chromatin from a distinct organism to every reaction. To this end, we added chicken chromatin (harvested from 50,000 untreated and unsynchronized DT-40 cells) to each ChIP-seq reaction. H3K9ac is expected to be enriched in promoters, enhancers, and insulators as we observed previously (Zhou et al, 2011; Javasky et al, 2018). As expected, the chicken chromatin presented a strong enrichment of H3K9ac at these genomic loci, thus validating our ChIP-seq conditions. We used the chicken promoter occupancy to normalize for the differences in ChIP efficiency between the samples (see the Materials and Methods section, Figs 5A and S4).

We used the normalized signal and compared H3K9ac occupancy at promoters, enhancers, and insulators in the different conditions

(Figs 5B and C and S5). In line with previous results by us and others (Javasky et al, 2018; Palozola et al, 2019), H3K9ac levels were high in unsynchronized (interphase) cells, whereas the acetylation declines during mitosis in untreated cells. Interestingly, all of the inhibitors we used had a similar impact on the levels of H3K9ac, with MS-275 (which inhibits both HDAC2 and HDAC3) having a slightly stronger effect when compared with CAY10683 (which inhibits only HDAC2) or RGFP966 (which inhibits only HDAC3). Of note, although the specific inhibition of HDACs in mitotic cells has demonstrated a strong increase in H3K9ac levels, the abundance of the signal was still lower than observed in unsynchronized, interphase cells. This potentially suggests that the regulation of histone deacetylation in mitosis involves the combination of multiple HDACs and a reduction in HAT activities.

## Discussion

Multiple studies have provided evidence for the presence of a histone deacetylation process during mitosis (Palozola et al, 2019). However, to our knowledge, the mechanism underlying this process has not been previously studied in detail. Here, we combined molecular and genomic techniques and an array of small molecules to identify the involvement of specific histone deacetylases during mitosis. Using this approach, we were able to detect three

regulators that are playing a key role in the mitosis-associated deacetylation of H3K9ac—HDAC2, HDAC3, and SIRT1 (Fig 2). In particular, we found that HDAC2 deacetylates H3K9ac in prophase and HDAC3 deacetylates this histone modification during metaphase. Although we observed that SIRT1 becomes detached from the chromatin during mitosis, we provide in vitro evidence that supports involvement of this sirtuin in impacting the levels of H3K9ac via repression of mitotic HAT activity.

Although inhibition of HDAC2 increases H3K9ac levels at prophase, the acetylation decreases again in metaphase, most probably because of the activity of HDAC3. Indeed, when we inhibited both HDAC2 and HDAC3, we observed an increase in H3K9ac levels throughout mitosis. On the other hand, HDAC3 inhibition did not affect H3K9ac levels at prophase (probably because of the activity of HDAC2) but induced an increase in H3K9ac levels at metaphase. By performing ChIP-seq on STC-arrested cells treated with various HDAC inhibitors, we identified the genomic locations of activity for each of these HDACs. We found that inhibition of either HDAC2 or HDAC3 has a similar effect on promoters, enhancers, and insulators (Fig 5C). Yet, there are certain genomic regions that appear to be preferably deacetylated by HDAC3 (Fig S6). These results are in line with our immunofluorescence results (Fig 2A) showing that HDAC3 has a stronger impact on H3K9ac levels during metaphase (most of the cells during an STC arrest are in metaphase).

Under normal growth conditions, increases in H3K9ac start at anaphase (Fig 1B). We observed that two HATs (HAT1 and EP300) remain associated with the mitotic chromosomes at all mitosis stages (Fig 3); however, HAT activity during metaphase appears to be reduced indirectly by SIRT1 (Fig 4A and B). This suggests that H3K9ac increases are delayed to anaphase because of the modulation of HAT activity by post translation modification (PTM). Indeed, SIRT1 inhibition causes an increase in H3K9ac levels already at metaphase (Fig 2). Modulation of HAT activity by PTMs is a well-documented phenomenon. Many of the HATs are activated by PTMs (Pavlik et al, 2013) and are frequently activated by auto-acetylation (McCullough & Marmorstein, 2016). Thus, it is reasonable to assume that HAT deacetylation may serve as a tool to partially repress the activity of key HATs during mitosis. Indeed, it was shown that EP300 is repressed by SIRT1-mediated deacetylation at lysine residues 1,020/1,024 (Bouras et al, 2005). Our observation is in line with these observations and suggests a physiological role for the regulation of HATs by SIRT1 at mitosis. However, inhibition of HDAC3 is associated with an increase in H3K9ac levels at metaphase (Fig 2), and this could potentially be because of a residual HAT activity.

Our observation that HAT1, a type B HAT that specifically acetylates free H4 at lysine residues 5 and 12, is localized to the mitotic chromosomes is interesting because historically this group of HATs was considered cytoplasmatic. Recent data suggest that HAT1 containing complexes are found in both the cytoplasm and the nucleus (Parthun, 2007). Moreover, during mitosis, because of the nuclear envelope break down, there is no more distinction between cytoplasm and nuclei. Further research is needed for characterizing the actual role of HAT1 during mitosis.

Our results suggest that the low levels of H3K9ac during mitosis are achieved by a combination of two HDACs and by the modulation of HAT activity by a third histone deacetylase. Indeed, it was shown

previously that the sole inhibition of class I and II HDACs by sodium butyrate is not sufficient to achieve full acetylation during mitosis because of the decreased ability of HATs to act on histones in mitotic chromatin (Patzlaff et al, 2010). It has been shown that RNA polymerase and many of the transcription factors fall off the mitotic chromatin (Martinez-Balbas et al, 1995; Parsons & Spencer, 1997; Kadauke & Blobel, 2013). Yet, the assumption that almost all transcription regulators are displaced from the chromatin during mitosis was recently challenged by a mass-spectrometry based study that showed a large-scale retention of TFs and preinitiation complex members on the mitotic chromatin (Ginno et al, 2018). Indeed, our results regarding the mitotic localization of several histone acetylases and deacetylases support the more complex picture of partial retention of transcription regulators on the mitotic chromatin (Fig 3). We found that HDAC1 and HDAC2 fall off the chromatin only at metaphase, whereas HDAC3 remains on the chromatin during all mitosis stages. The sirtuins show a partial chromatin association at prophase, and they are not seen at the chromatin from metaphase on. Finally, we found that some of the HATs (EP300 and HAT1) remain on the chromatin, whereas others (CBP and GCN5L2) fall off. These results are only partially consistent with previous observations. The displacement of HDAC1 and HDAC2 from metaphase through the rest of mitosis stages was reported previously in MCF7 cells (He et al, 2013). On the other hand, HDAC3 was found on mitotic chromatin in HeLa-S3 cells (Li et al, 2006) but not on MEFs (Bhaskara et al, 2008). Similarly, SIRT1 was found by us to evict the mitotic chromosomes at the metaphase stage in HeLa-S3 cells (Fig 3), whereas in MEFs, it seems to be retained on the chromatin along all mitotic stages (Fatoba & Okorokov, 2011). These discrepancies are either because of the use of different techniques or because of the use of different experimental systems. Further research is required to determine whether the conflicting results reflect the above differences or variations between primary and transformed cells. Our conclusions are based on the use of small-molecule HDACs inhibitors that target specific HDACs or groups of histone deacetylases (Table 1). To mitigate the dependency on each of the small molecules' specificity, we aimed to employ multiple inhibitors to target key deacetylases. Thus, we evaluated the involvement of HDAC2 by using MS-275, chidamide, and CAY10683 and studied HDAC3 by using RGFP966, chidamide, and MS-275 (Table 1). However, for SIRT1, we could only use EX-527, and thus, we currently cannot rule out the involvement of additional sirtuins in the process, and further research is required to study this possibility.

Based on immunofluorescence and Western blot analyses (Fig 2), we deduced that HDAC2 and SIRT1 are released from the mitotic chromosomes during metaphase. Previous studies (Pallier et al, 2003; Teves et al, 2016; Festuccia et al, 2019) suggest that PFA fixation may artificially cause such eviction. To evaluate this possibility, we repeated the IF experiments of SIRT1 and HDAC2 using the DSG cross-linker in addition to PFA because it was suggested that such cross-linking is more suitable for studying mitotic retention of proteins (Festuccia et al, 2019). In both cases, we found similar results with both fixation approaches (Fig S7). These results suggest that SIRT1 and HDAC2 are resistance to the potential impact of PFA on mitotic chromatin retention. These observations are also consistent with the mild changes in H3K9ac levels upon inhibiting HDAC2 at metaphase (Fig 2). However, it should be noted that these

experiments were performed in fixed cells and are thus limited. Future work using live cell imaging of fluorescently labeled proteins would be required to provide additional support for the localization patterns of HDACs in normal unperturbed growth conditions.

Taken together, our results suggest a complex mechanism for H3K9 deacetylation during mitosis. The deacetylation of H3K9 is carried out by a combination of HDACs that are activated at different mitotic stages and potentially via a reduction in HAT activity that is modulated by SIRT1. In addition, we identified a temporal separation in the activity of the HDACs, with HDAC2 acting at prophase on most transcription-associated H3K9ac peaks, whereas HDAC3 contributes at a later stage.

Last, we note that further research is needed to better understand several additional aspects of HDAC2 and HDAC3 activity during mitosis. For instance, in past work, we observed that many nucleosome depletion regions are specifically deacetylated during mitosis (Javasky et al, 2018). It would be interesting to see if one or both HDACs are playing a role in this targeted deacetylation of a single nucleosome. In addition, what is the genomic localization of HDAC2 and HDAC3, and does it change during the phases of mitosis? Furthermore, what is the impact of each wave of deacetylation on mitotic traits such as transcription repression and chromosome condensation?

## Materials and Methods

### Cell culture and cell cycle synchronization

HeLa-S3 cells and MEFs were grown in DMEM supplemented with 10% FBS, penicillin–streptomycin, L-glutamine, sodium pyruvate, and 0.1% Pluronic F-68. Cells were incubated at 37°C and 5% $CO_2$. DT40 cells were cultured in RPMI 1640 medium supplemented with 10% fetal calf serum, 1% chicken serum, 10 mM Hepes, and 1% penicillin–streptomycin mixture at 39.5°C with 5% $CO_2$.

For STC mitotic arrest, cells were pre-synchronized in G1/S by addition of 2 mM thymidine for 16 h, washed with PBS, released for 3 h in fresh medium, and arrested with 10 $\mu$M S-trityl-L-cysteine (STC) (164739; Sigma-Aldrich) for 12 h. For double thymidine synchronization, HeLa-S3 cells were grown on coverslips in DMEM supplemented with 10% FBS, penicillin–streptomycin, L-glutamine, sodium pyruvate, and 0.1% Pluronic F-68. Cells were incubated at 37°C and 5% $CO_2$. Cells were grown for 20 h, treated with 2 mM thymidine for 18 h, washed with PBS, released for 8 h in fresh medium, treated with 2 mM thymidine for 17 h, washed with PBS, released for 8.5 h in fresh medium, and harvested.

### HDAC and sirtuin inhibitors

The concentrations and specificities of the inhibitors we used are summarized in Table 1. The inhibitors were added for 8.5–9 h (Fig 1A and C).

### Chromatin isolation

Chromatin was isolated from STC-synchronized HeLa-S3 cell line using a previously published protocol (Sone et al, 2002). Synchronized cells were collected by centrifugation and resuspended into a hypotonic solution of 75 mM KCL (pH 5.7). After treatment of KCL hypotonic solution for 30 min, the cells were collected by centrifugation and resuspended into CAS buffer (0.1 M citric acid, 0.1 M sucrose, 0.5% Tween 20, pH 2.6). After lysis of the cell membrane in CAS buffer, the chromatin suspension was centrifuged at 190$g$ for 3 min at 4°C. The chromatin-rich fraction was carefully recovered as supernatant, and the nuclei rich fraction was recovered as the precipitation. The supernatant fraction was then centrifuged again at 1,750$g$ for 10 min at 4°C. The precipitated chromatin was resuspended in CAS buffer containing 0.1 mM phenylmethylsulfonyl fluoride. The isolated mitotic chromatin quantity was estimated by BCA method and stored in −80°C until further use.

Chromatin from asynchronous cells was isolated as described (Torrente et al, 2011). Cells were resuspended in Buffer A (10 mM Hepes [pH = 7.9], 10 mM KCl, 1.5 mM $MgCl_2$, 0.34 M sucrose, 10% glycerol, inhibitor cocktail: 1 mM DTT, 0.5 mM 4-(2-aminoethyl) benzenesulfonyl fluoride hydrochloride, and protease inhibitor cocktail). Triton X-100 was added to a final concentration of 0.1%, and the suspension was incubated for 8 min on ice. The nuclear pellet was obtained by centrifugation (1,300$g$ for 5 min at 4°C), washed with Buffer A, and then resuspended in Buffer B (3 mM EDTA, 0.2 mM EGTA, and protease inhibitor cocktail) for 30 min on ice. The insoluble chromatin pellet was isolated by centrifugation (1,700$g$ for 5 min at 4°C) and then resuspended in 15 mM Tris, pH = 7.5, 0.5% SDS. The isolated mitotic chromatin quantity was estimated by the BCA method and stored in −80°C until further use.

### Immunoblotting

Isolated chromosomes were directly suspended and dissolved in SDS sample buffer (62.5 mM Tris–HCl [pH 6.8], 5% 2-mercaptoethanol, 20% glycerol, 2% SDS, and 0.005% bromophenol blue). Purified chromatin was separated by SDS–PAGE and transferred to 0.45-$\mu$m polyvinylidene difluoride membranes (Immobilon, Millipore, Merck KGaA). Blots were incubated with primary antibodies HDAC1 (#D5C6U; 1; 1,000; Cell Signalling), HDAC2 (#D6S5P; 1:1,000; Cell Signalling), HDAC3 (#7G6C5; 1; 1,000; Cell Signalling), CBP (#D6C5; 1; 1,000; Cell Signalling), GCN5L2 (#C26A10; 1:1,000; Cell Signalling), P300 (#sc-48343; 1:100; Santa Cruz), HAT-1 (# sc-390562; 1:1,000; Santa Cruz), SIRT1 (#8469; 1; 1,000; Cell Signalling), SIRT2 (#sc-28298; 1:1,000; Santa cruz), SIRT6 (#ab62739; 1:1,000; Abcam), SIRT7 (#135055; 1:1,000; Santa Cruz), and histone H3 acetyl K9 (Ac-H3K9; [#C5B11; 1:1,000; Cell Signalling]). HRP-conjugated secondary antibodies (#111-035-003 and #115-035-003; 1:5,000; Jackson Immuno Research) were used. Immunoblots were developed with an ECL-plus kit. Equal loading of protein in each lane was verified by histone H3 (#D1H2; 1:1,000; Cell Signalling).

### Immunofluorescence staining

For immunofluorescence studies, HeLa-S3 cells and MEFs were grown on coverslips, fixed with 4% formaldehyde (Cat. no. 28908; Thermo Fisher Scientific). Alternatively, HeLa-S3 cells were cross-linked with 2 mM DSG (Cat. no. 80424; Sigma-Aldrich) for 50 min followed by 10 min incubation with 1% formaldehyde at RT (Festuccia et al, 2019). After fixation, cells were blocked in PBS

containing 5% BSA and 0.1% Triton X-100. Cells were then incubated with a primary antibody—HDAC1 (#D5C6U; 1:100; Cell Signalling), HDAC2 (# D6S5P; 1:1,000; Cell Signalling), HDAC3 (#7G6C5; 1:100; Cell Signalling), CBP (#D6C5; 1:100; Cell Signalling), GCN5L2 (#C26A10; 1:100; Cell Signalling), P300 (#sc-48343; 1:100; Santa Cruz), HAT-1 (# sc-390562; 1:100; Santa Cruz), SIRT1 (#8469; 1; 100; Cell Signalling), SIRT2 (#sc-28298; 1:100; Santa Cruz), SIRT6 (# ab62739; 1:100; Abcam), SIRT7 (#135055; 1:100; Santa Cruz), and H3K9ac (#C5B11; 1:400; Cell Signalling) overnight at 4°C. After washing, cells were incubated with an anti-rabbit IgG or anti-mouse IgG conjugated to FITC probes (#A11034 and #A11001; Invitrogen) for 1 h at RT. After washing, slides were mounted with SlowFade Gold antifade reagent with DAPI (Sigma-Aldrich), and immunofluorescent signals were viewed using an Olympus fluorescence microscope using CellSens software. All images were taken under fixed scaling and were normalized to only secondary background control to avoid false significance because of overexposure. Images were analyzed with ImageJ (Fiji version) to quantify mean intensity/pixel in the DAPI stained regions.

### In vitro HAT activity assay

In vitro histone acetylation assays were performed on 25 $\mu$g of mitotic chromatin (isolated as described above and quantified using the bicinchoninic acid colorimetric assay system [Sigma-Aldrich]) in 50 mM Tris–HCL, 10% glycerol, 1 mM DTT, 1 mM PMSF, 50 nM TSA, 0.1 mM EDTA supplemented with 100 $\mu$M Acetyl-CoA, and 10 $\mu$g His-tagged histone H3 (#52023; BPS Bioscience). The reaction was incubated at 37°C for 1 h. HAT activity was quantified by: (i) a colorimetric assay quantifying the release of Co-A using DTNB (2-nitrobenzoic acid) (Sigma-Aldrich) by measuring the absorbance at 412 nm; (ii) isolating the His-tagged H3 by Ni-NTA column and measuring H3K9ac by immunoblots using anti-H3K9Ac antibody (Cell Signalling #C5B11; 1:1,000). All acetylation assays were performed in duplicate.

### In vitro HDAC3 activity assay

Mitotic HeLa-S3 cells were lysed in cell lysis buffer (0.5% NP-40, 20 mM Tris–HCl [pH 7.4], 500 mM NaCl, 0.5 mM EGTA, 10% glycerol, 0.5% Triton-X 100, complete protease inhibitor cocktail and 1 $\mu$M of zinc citrate). Mitotic lysate was precleared by the protein A magnetic bead (Cell Signalling) for 20 min at RT. After protein quantification by the bicinchoninic acid colorimetric assay system (Sigma-Aldrich), 1 mg protein samples were used for HDAC3 immunoprecipitation using 4 $\mu$l of HDAC3 antibodies (Cell Signalling) and incubated at 4°C overnight, followed by 2 h incubation with 20 $\mu$l protein A magnetic beads. The captured beads were rinsed with lysis buffer five times and (i) boiled in 5× SDS loading buffer for 5 min for confirming HDAC3 immunoprecipitation by immunoblot (after brief centrifugation, supernatants were loaded onto 12% SDS–PAGE gels and detected using rabbit-anti-HDAC3 antibody [1:1,000; Cell Signalling] and HRP-conjugated secondary antibodies), or (ii) immunocomplex was eluted from bead in 0.2 M glycine (pH 2.6) by incubating the sample for 10 min and neutralized by adding equal volume of Tris–HCL (pH 8.0), and stored at –20°C for further use.

To perform HDAC activity assay, we used 50 ng of immunoprecipitated HDAC3 and 4 $\mu$g of His-tagged H3 (BPS Bioscience #52023; acetylated by pre-treatment with whole cell extract and isolated by Ni-NTA column), in HDAC Assay buffer (10 mM Tris–HCl [pH 8.0], 10 mM NaCl, 1 $\mu$M ZnSO$_4$ 10% glycerol, and complete protease inhibitor mixture), incubated at 37°C for 1 h. After the incubation, SDS–PAGE loading buffer is added to the reaction and runs on 12% SDS–PAGE. HDAC3 activity in the presence or absence of inhibitors was assessed by immunoblotting using anti H3K9ac antibody (#C5B11; 1:1,000; Cell Signalling).

### ChIP-seq

ChIP experiments were carried out as described in Mikkelsen et al (2007) and Texari et al (2021). All ChIP experiments were carried out on 2 × 10$^5$ cells. 1.5 × 10$^5$ HeLa-S3 cells (unsynchronized or STC synchronized with or without the indicated treatments) and 0.5 × 10$^5$ unsynchronized and untreated DT-40 cells were cross-linked with 4% formaldehyde and mixed. The cell mixture was lysed and sonicated by Covaris to obtain an average chromatin size of 200–700 bp. Immunoprecipitation, using anti-H3K9ac (C5B11; Cell Signalling Technology), was carried out with inverting at 4°C for 14–16 h. Antibody–chromatin complexes were pulled down using Protein A/G magnetic beads (Dy-10001D and Dy-10003D; Invitrogen), washed, and then eluted. After cross-link reversal and Proteinase K treatment, immunoprecipitated DNA was extracted with 1.8× Agencourt AMPure XP beads (BCA63881; Beckman Coulter).

### Library preparation and sequencing

ChIP DNA (or unenriched whole cell extract) were prepared for Illumina sequencing as described (Yehuda et al, 2018). Briefly, DNA was subjected to a 50 $\mu$l end repair reaction, cleaned by 1.8× AMPure XP beads, followed by a 50 $\mu$l A-tail reaction. The products were cleaned and ligated to 0.75 $\mu$M Illumina compatible forked indexed adapters. Ligation products were size selected to remove free adapters. Ligation products were amplified with 15 (Input DNA) or 18 PCR cycles (ChIP DNA). Amplified DNA was size selected for 300–700 bp fragments using AMPure XP beads. The final quality of the library was assessed by Qubit and TapeStation. Libraries were pooled and sequenced on NextSeq (Illumina), using a paired-end protocol. The sequencing depth of each library is provided at Table S1.

### Bioinformatic analysis

ChIP-seq reads were aligned to the human (hg38) or chicken (galGal6) genomes using Bowtie2 (Langmead & Salzberg, 2012). All regions listed in the ENCODE hg38 blacklist were excluded from all analyses. Duplicate alignments were removed with Picard Tools MarkDuplicates (http://broadinstitute.github.io/picard/). Only reads with mapping quality >30 were used in the analysis. Peak detection on merged unsynchronized replicates (with a whole cell extract dataset as a control) was done using MACS2 (Zhang et al, 2008) with default parameters and q-value of 0.05.

Metagene analyses were done using the python package Metaseq (Dale et al, 2014). The metagenes were aligned to human transcription start sites (n = 60,580, taken form ENSEMBL GTF files),

enhancers, and insulators (n = 4,274 and 2,025, respectively, taken from Javasky et al [2018]); and to chicken TSS (n = 7,234, taken form ENSEMBL GTF files), enhancers (n = 20,633, retrieved from the enhancer atlas [Gao & Qian, 2020]), and insulators (n = 13,510 CTCF binding sites, retrieved from GSM1253767). IGV snapshots show TDF files created from the aligned sequence data (using the count command from IGV tools [Robinson et al, 2011; Thorvaldsdottir et al, 2013]). HOMER annotatePeaks.pl was used to quantify sample coverage at each peak, shown in Fig S6. Quantification of read coverage for all analyses in all datasets was normalized to the total number of reads multiplied by a normalization factor derived from the chicken promoter data.

## Data Availability

The Chip-Seq datasets produced in this study are available in Gene Expression Omnibus GSE168180.

## Supplementary Information

## Acknowledgements

We thank the members of the Core Research Facility in the Hebrew University School of Medicine; Dr. Idit Shiff and Dr. Abed Nasereddin for generating genomic data. This work was supported by research grants from the Israel Science Foundation (grant # 1283/21 to I Simon), ISF-NSFC (grant #2555/16 to I Simon), the Binational Science Foundation (joint program with NSF; grant # 2019688), and NSF/MCB-BSF (grant # 2003358 to A Goren).

### Author Contributions

S Gandhi: conceptualization, formal analysis, investigation, visualization, methodology, and writing—original draft.
R Mitterhoff: resources, software, formal analysis, investigation, and methodology.
R Rapoport: resources, data curation, software, and formal analysis.
M Farago: data curation, validation, investigation, visualization, and methodology.
A Greenberg: investigation and visualization.
L Hodge: data curation, software, and formal analysis.
S Eden: conceptualization, visualization, methodology, and writing—review and editing.
C Benner: data curation, software, and supervision.
A Goren: conceptualization, data curation, supervision, funding acquisition, project administration, and writing—original draft, review, and editing.
I Simon: conceptualization, data curation, formal analysis, supervision, funding acquisition, project administration, and writing—original draft, review, and editing.

## Conflict of Interest Statement

The authors declare that they have no conflict of interest.

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
