## [Reviewer comments · Life Science Alliance]

Life Science Alliance

Mitotic H3K9ac is controlled by phase-specific activity of HDAC2, HDAC3 and SIRT1

Shashi Gandhi, Raizy Mitterhoff, Rachel Rapoport, Marganit Farago, Avraham Greenberg, Lauren Hodge, Sharon Eden, Christopher Benner, Alon Goren, and Itamar Simon

DOI: <https://doi.org/10.26508/lsa.202201433>

Corresponding author(s): *Itamar Simon, Hebrew University of Jerusalem*

Review Timeline:

Submission Date:	2022-03-04
Editorial Decision:	2022-03-29
Revision Received:	2022-07-04
Editorial Decision:	2022-07-26
Revision Received:	2022-07-29
Accepted:	2022-08-01

Scientific Editor: Novella Guidi

Transaction Report:

March 29, 2022

Re: Life Science Alliance manuscript #LSA-2022-01433-T

Prof. Itamar Simon
Hebrew University of Jerusalem
Microbiology and Molecular Genetics
Hadassah Ein Kerem
Jerusalem, Please select: 91101
Israel

Dear Dr. Simon,

Thank you for submitting your manuscript entitled "Mitotic H3K9ac is controlled by phase-specific activity of HDAC2, HDAC3 and SIRT1" to Life Science Alliance. The manuscript was assessed by expert reviewers, whose comments are appended to this letter. We, thus, encourage you to submit a revised version of the manuscript back to LSA that responds to all of the reviewers' points.

Thank you for this interesting contribution to Life Science Alliance. We are looking forward to receiving your revised manuscript.

Sincerely,

B. MANUSCRIPT ORGANIZATION AND FORMATTING:

Reviewer #1 (Comments to the Authors (Required)):

In this manuscript, Gandhi et al describe how the H3K9ac levels are controlled by different HDAC enzymes at distinct stages of mitosis. Using mostly Immunofluorescence, the authors detect a gradual decrease of H3K9ac levels during prophase and metaphase. Then, they use multiple HDAC inhibitors on synchronized cells (exiting S phase) to determine which ones are responsible for de-acetylation and discover a stage-specific involvement of different HDACs. The observations are interesting especially under the light of recent studies that nominate histone acetyl residues as potential mitotic bookmarks at select genomic regions (although the authors do not cite this work). There are several technical issues that need to be addressed:

- All IF quantitations should be presented as boxplots (or violin/diamond plots), indicating the number of nuclei that were measured, the range/variability of signal and the statistical differences. Overall statistical analysis is very limited or absent in most comparisons.
- To build a more convincing case for the differential effect of each inhibitor during prophase/metaphase, the authors need to show on asynchronous cells the efficiency and kinetics of the inhibitors that they use. Do all inhibitors work efficiently during an 8hour treatment? What are the levels of remaining histone acetylation (K9ac and global acetyl) during and after the treatments? The 8hour window is a very extensive treatment, and it would be beneficial to determine the minimum effective window to avoid secondary effects.
- Also, -at least- a simple cell cycle analysis following each of the treatments would be informative for the role of each of these HDACs on cell cycle progression. Do they mess up the ability of cells to exit mitosis?
- A lot of literature is not properly discussed and cited. There is literature showing that IF is not ideal for capturing mitotic retention of protein factors (Teves et al, Festuccia et al etc). There are many papers showing partial retention of histone acetylation marks during mitosis either by mass-spec and/or ChIP-seq (Behera et al 2019, Kang et al 2020, Liu et al 2017, Pelham-Webb et al 2021). Some of them suggest functional bookmarking activity for the retained acetyl at select genomic regions. The authors should carefully discuss their findings acknowledging prior work.
- The ChIP-seq analysis could be interesting, if the authors look deeper into the relative specificity of each HDAC inhibitor. Do the treatments simply affect the global H3K9ac levels or do they preferentially affect only a subset of H3K9ac loci.
- Several typos throughout the text (e.g enhancers instead of enhancers etc)

Reviewer #2 (Comments to the Authors (Required)):

In this paper, the authors dissect the contribution of HDACs to the loss of H3K9ac during mitosis. They find that specific activities play temporally distinct roles (HDAC2 in prophase; HDAC3/SIRT1 in metaphase). They also suggest that only HDAC3 is strongly associated with mitotic chromatin (metaphase) and propose that the role of SIRT1 is not direct but mediated by inhibiting HAT activity. Finally, they perform quantitative ChIP-seq to show that inhibiting HDACs leads to partial restoration of H3K9ac in mitotic cells. Overall, the paper provides a thorough and well conducted analysis of HDAC function during mitosis, focused on H3K9ac, which will be useful for the field. I have a number of questions/comments:

1/ The mitotic retention of chromatin regulators is strongly influenced by experimental conditions, particularly fixation. I could not find in the methods how cells were fixed for IF, but I suppose it was PFA. If so, and as the authors likely know, PFA artificially depletes diffusible proteins from mitotic chromosomes. I think it would be very important to validate their observations of exclusion of HDACs and HATs with a) GFP fusion proteins and b) alternative fixative conditions such as DSG or Glyoxal, as my lab reported (Festuccia et al. Genome Research 2019).

2/ It is unclear to me whether the impact of Sirt1 on HAT activity is specific to mitosis. Did the authors check asynchronous cells? I think it would be an important addition. Similarly, can the authors clarify if they think Sirt1 interacts with HDACs? Performing coIP assays with Hdac3 would seem very relevant.

3/ If the authors produce more convincing evidence of the exclusion of all HATs from mitotic chromatin except p300 and HAT1, I would like them to discuss the fact that HAT1 is principally acetylating H4 in the cytoplasm. Also, their views of how is H3K9 acetylated upon HDAC inhibition if main H3K9 acetyl-transferases are evicted from mitotic chromatin.

I hope the authors will find these comments useful
Pablo Navarro

Reviewer #3 (Comments to the Authors (Required)):

The authors of this MS provide a model of mitotic regulation of H3K9ac in which H3K9 deacetylation is a stepwise process - at prophase HDAC2 modulates most transcription-associated H3K9ac-marked loci and at metaphase HDAC3 maintains the reduced acetylation, whereas SIRT1 potentially regulates H3K9ac by impacting HAT activity. Although the results presented in the MS are of potential interest and this MS provides a characteristic pattern about regulation of H3K9ac from extensive data, there are several technical issues which require to be addressed before accepted for publication in Life Science Alliance.

For main point 1 : Identification of the deacetylases involved in modulating H3K9ac during mitosis.

The major issue: The identification of histone deacetylases modulating H3K9ac during mitosis using only specific inhibitory treatments is insufficient, and it is necessary to construct knockdown cell lines for the screened HDAC2, HDAC3 and Sirt1 to demonstrate that mitotic H3K9ac is controlled by phase-specific activity of HDAC2, HDAC3 and SIRT1. The timeframe required is about four months.

For main point 2 : Variations in the mitotic chromatin association of key histone deacetylases and histone acetyltransferases.

The major issue: Identification of the chromatin localization patterns of key HATs and HDACs during mitosis. Further immunofluorescence co-localization analysis of HDAC3, HAT1 and EP300 signals detected in metaphase is required to explain variations in the mitotic chromatin association of key histone deacetylases and histone acetyltransferases. The timeframe required is one month.

For main point 3 : SIRT1 potentially impacts H3K9ac levels indirectly by modulating HAT activity.

The major issue: p300 and HAT-1 were the acetylase, which were both detected on the metaphase chromatin. Experiments about these enzymes with the same treatment as Fig5C are needed to further confirm that SIRT1 indirectly affects H3K9ac levels by potentially modulating specific HAT activity during mitosis. The timeframe required is two months.

For main point 4 : HDAC2 and HDAC3 activity regulates both common and unique genomic loci.

The quality of ChIP-seq data provided is acceptable.

Other issues:

a: The scale of the picture is not marked.

b: When measured H3K9ac levels in cells captured at different stages of mitosis, the cell sample size is only 20 cells and should be at least more than 100 cells in order to reduce the error of quantification of the immunofluorescence results.

c: Although it is mentioned in the manuscript that the significant difference analysis is based on $P < 10^{-23}$; one sided t test, the degree of significance should be marked with * in the figure.

d: There are subscript errors in the drug name and chemical composition in the manuscript, such as line 417, 421, which need to be corrected.

f: Exposure parameters for immunofluorescence microscopy need to be provided to avoid false significance due to overexposure.

g: High-resolution, undistorted images are required, especially the cytology images in Figure 2.

To the editor:

We would like to thank the reviewers for their careful reading of our manuscript and for their insightful comments. Overall, we were encouraged by the reviewers' enthusiasm for our manuscript and its value to the community.

We address each of the issues raised by the reviewers in detail in point-by-point responses below. Reviewers' comments are presented in italics and dark blue, our responses in standard font.

Reviewer #1:

In this manuscript, Gandhi et al describe how the H3K9ac levels are controlled by different HDAC enzymes at distinct stages of mitosis. Using mostly Immunofluorescence, the authors detect a gradual decrease of H3K9ac levels during prophase and metaphase. Then, they use multiple HDAC inhibitors on synchronized cells (exiting S phase) to determine which ones are responsible for de-acetylation and discover a stage-specific involvement of different HDACs.

The observations are interesting especially under the light of recent studies that nominate histone acetyl residues as potential mitotic bookmarks at select genomic regions (although the authors do not cite this work).

We thank the reviewer for appreciating the relevance of our work and for noting the missing citations. We added the missing reference in the second paragraph of the introduction.

There are several technical issues that need to be addressed:

- All IF quantitations should be presented as boxplots (or violin/diamond plots), indicating the number of nuclei that were measured, the range/variability of signal and the statistical differences. Overall statistical analysis is very limited or absent in most comparisons.

We thank the reviewer for noting these points. Apart from the exception below, we have replaced all the bar graphs in the figures with boxplots. We included the number of observations and the statistical tests used in the figure captions.

For **Figure 2A** and **2B**, we kept the bar graphs and added a box plot representation of the data as a supplementary figure (**Supplementary Figure S3**; also added below as **Fig. 1**). We preferred this option since bar graphs allow clearer representation of the high number of datapoints we have.

Fig. 1. Box plot representation of the results of Figure 2A and 2B. The different conditions are depicted as colors and presented on the right.

- To build a more convincing case for the differential effect of each inhibitor during prophase/metaphase, the authors need to show on asynchronous cells the efficiency and kinetics of the inhibitors that they use. Do all inhibitors work efficiently during an 8hour treatment? What are the levels of remaining histone

acetylation (K9ac and global acetyl) during and after the treatments? The 8hour window is a very extensive treatment, and it would be beneficial to determine the minimum effective window to avoid secondary effects.

We thank the reviewer for these suggestions. In interphase, the inhibitors showed only a minor effect (**Figure 2 of the paper**). Thus, we conducted the kinetic experiments suggested by the reviewer on cells arrested in metaphase by a double thymidine block. Interphase cells were treated for 2, 4, 6 and 8 hours with the key inhibitors used in our study (namely, RGFP966, CAY10683 and MS-275) and the effect on H3K9ac was measured by IF. The measurements during these timepoints are in tight accordance with our findings, across all time points (**Fig. 2**). Further, this analysis demonstrates that cells treated by the inhibitors for shorter time periods presented a lower reduction in H3K9ac, thus confirming the need to use the longer incubation time we used. Further, FACS analysis of the cell cycle distribution following treatment for 8 hours has shown that the cell viability was mainly unimpacted.

We agree with the reviewer that studying the changes in global histone acetylation levels is of interest. Yet, as our study is focused on the changes in H3K9ac and not global acetylation levels and the high work burden required for including another antibody for these multiple experiments, we envision this to be beyond the scope of this report.

Fig. 2. Evaluation of the impact of length of HDACi treatments on H3K9ac levels on metaphase cells. Mean intensity per pixel and standard errors of H3K9ac at the metaphase stage for different durations of treatment of the indicated HDACi. The data were normalized to the intensity of untreated (NT) cells (few data points are missing). Note, that the basic findings, namely the difference between CAY10683, which have a small effect on metaphase cells, and RGFP966, which show a much stronger effect, is similar regardless of the duration of the treatment. The effect increases with treatment time and is most obvious at the 8-hour time point. The results are based on >20 measurements at each time point.

- Also, -at least- a simple cell cycle analysis following each of the treatments would be informative for the role of each of these HDACs on cell cycle progression. Do they mess up the ability of cells to exit mitosis?

As the reviewer suggested, we performed FACS analysis to demonstrate that the 8-hour treatment does not have a strong impact on the cell cycle, and thus does not seem to harm the cells (**Fig. 3**)

Manual Model	G1%	S%	G2%
Nt	52.87	22.63	13.18
Cay	45.15	22.40	19.92
Ex	49.91	19.52	21.13
Ms	46.22	22.10	20.47
Rg	53.65	16.26	21.38
TSA	50.12	15.55	22.72

Fig. 3. Cell cycle analysis of cells treated for 8 hours with HDACi. Histograms showing DNA content of cells treated for 8 hours with the indicated HDACs inhibitors. Note, that the treated cells show a typical cell cycle with a slight increase of G2/M cells.

In addition, we looked directly at the ability of cells to exit mitosis under the various treatments. We observed that while the pan-HDAC inhibitor TSA prevents the cells from exiting the mitosis phase, the other inhibitors we used (HDAC specific) did not have this effect (**Fig. 4**).

Fig. 4. Selective HDAC inhibitors do not prevent G1 reentry. Quantification (using ImageJ) of the number of cells in interphase at 4 timepoints following the release from STC mitotic arrest and treatment with the HDAC inhibitors TSA, Chidamide, Nicotinamide and RGFP966. While cells treated with the global non-selective HDAC inhibitor TSA do not reenter G1, cells treated with the HDAC specific inhibitors appear to reenter G1 similarly to untreated cells 6 hours after release from mitotic arrest.

- A lot of literature is not properly discussed and cited. There is literature showing that IF is not ideal for capturing mitotic retention of protein factors (Teves et al, Festuccia et al etc).

We thank the reviewer for these suggestions and apologize for the omission. We have now added a paragraph in the Discussion in which we consider these papers. Further, we conducted an experiment with DSG-based fixation to evaluate the possibility that SIRT1 and HDAC2 eviction stems from the FA fixation (**Supplementary Figure S7**). We observed the same patterns of SIRT1 and HDAC2 eviction from mitotic chromatin under these conditions as well, supporting our initial observation using FA.

There are many papers showing partial retention of histone acetylation marks during mitosis either by mass-spec and/or ChIP-seq (Behera et al 2019, Kang et al 2020, Liu et al 2017, Pelham-Webb et al 2021). Some of them suggest functional bookmarking activity for the retained acetyl at select genomic regions. The authors should carefully discuss their findings acknowledging prior work.

This is indeed correct and was mainly discussed in our previous paper (Javasky et al., Genome Research, 2018). The focus of the current paper is on the mechanism of the deacetylation and not about the function of the changes in acetylation patterns. We added the missing references in the second paragraph of the introduction.

- The ChIP-seq analysis could be interesting, if the authors look deeper into the relative specificity of each HDAC inhibitor. Do the treatments simply affect the global H3K9ac levels or do they preferentially affect only a subset of H3K9ac loci.

We have performed the suggested analysis, and tested these loci for GO annotations, proximity to genes and ncRNAs but were not able to find any major differences between the genomic regions affected specifically by each inhibitor.

- Several typos throughout the text (e.g enhancers instead of enhancers etc).

We apologize for the errors and we carefully proofread the manuscript to ensure that there are no typos.

Reviewer #2:

In this paper, the authors dissect the contribution of HDACs to the loss of H3K9ac during mitosis. They find that specific activities play temporally distinct roles (HDAC2 in prophase; HDAC3/SIRT1 in metaphase). They also suggest that only HDAC3 is strongly associated with mitotic chromatin (metaphase) and propose that the role of SIRT1 is not direct but mediated by inhibiting HAT activity. Finally, they perform quantitative ChIP-seq to show that inhibiting HDACs leads to partial restoration of H3K9ac in mitotic cells. Overall, the paper provides a thorough and well conducted analysis of HDAC function during mitosis, focused on H3K9ac, which will be useful for the field.

We thank the reviewer for the encouraging remarks regarding the usefulness of our study.

I have a number of questions/comments:

1/ The mitotic retention of chromatin regulators is strongly influenced by experimental conditions, particularly fixation. I could not find in the methods how cells were fixed for IF, but I suppose it was PFA. If so, and as the authors likely know, PFA artificially depletes diffusible proteins from mitotic chromosomes. I think it would be very important to validate their observations of exclusion of HDACs and HATs with a) GFP fusion proteins and b) alternative fixative conditions such as DSG or Glyoxal, as my lab reported (Festuccia et al. Genome Research 2019).

We apologize for not clearly describing in the methods our fixation procedure, and fixed it in the revised manuscript. We used 4% formaldehyde and agree that it would be valuable to repeat our key observations using alternative fixation conditions.

We thank the reviewer for the suggestion to use DSG crosslinking for evaluating the eviction of SIRT1 and HDAC2 from the chromosomes at Metaphase. Following their suggestion, we repeated those experiments using a double fixation protocol (DSG and formaldehyde as described in (Festuccia et al.,

2019)). The results were in accordance with original observations, and SIRT1 was evicted from chromatin at metaphase in the double fixed cells as well.

The reviewer suggestion allowed us to identify that the association of HDAC2 with the chromosomes during metaphase is maintained at low levels (**Supplementary Figure S7**). This was also observed using our original FA fixation (**Figure 3**). We modified the text (**lines 208-211**) to account for this better interpretation. These low levels can be detected only by IF and not by western blot (Original **Figure 3d**). Moreover, this is consistent with the mild changes in H3K9ac levels upon inhibiting HDAC2 at metaphase, compared with the strong effect of HDAC3 inhibition (**Figure 2**). We added a paragraph in the Discussion about the effect of different fixation methods (**lines 409-419**).

We agree that using a GFP fusion proteins can provide additional support to our observations. Yet, it is beyond the scope of this study, as in our hands, creating the constructs and optimizing conditions usually takes at least 4-6 months.

2/ It is unclear to me whether the impact of Sirt1 on HAT activity is specific to mitosis. Did the authors check asynchronous cells? I think it would be an important addition. Similarly, can the authors clarify if they think Sirt1 interacts with HDACs? Performing coIP assays with Hdac3 would seem very relevant.

While we agree with the reviewer that these are intriguing questions, they are not related to the specific questions we were aiming to address. In particular, we observed that even though SIRT1 is not associated with mitotic chromatin, specific inhibition of this deacetylase increases mitotic histone acetylation. We show that a potential mechanism for this observation is that SIRT1 impacts the deacetylation of H3K9 indirectly during mitosis. While SIRT1 may potentially have a similar indirect activity during interphase, we do not envision that addressing this question is relevant to our study. Further, while coIP may provide additional information, studying the interactions of SIRT1 is beyond the scope of our study. We did not expect that SIRT1 and HDAC3 interact, and while we consider that SIRT1 is potentially deacetylating HATs we do not know which specific HATs are the targets of this deacetylase and it thus would require too large of a survey to identify the target HAT/s.

3/ If the authors produce more convincing evidence of the exclusion of all HATs from mitotic chromatin except p300 and HAT1, I would like them to discuss the fact that HAT1 is principally acetylating H4 in the cytoplasm. Also, their views of how is H3K9 acetylated upon HDAC inhibition if main H3K9 acetyl-transferases are evicted from mitotic chromatin.

We thank the reviewer for this suggestion. Adding additional evidence for the occupancy of the various HAT proteins on the mitotic chromosomes is out of the scope of the current manuscript.

The question regarding the HAT that acetylates H3K9 when the HDACs are inhibited is indeed intriguing, and requires additional work. We consider that in spite of the down regulation of HAT activity by SIRT1, there is still residual HAT activity that is sufficient to increase H3K9ac levels when the relevant HDACs are inhibited.

I hope the authors will find these comments useful
Pablo Navarro

Dear Dr. Navarro – we appreciate the time and effort you put in reading our manuscript and providing suggestions!

Reviewer #3:

The authors of this MS provide a model of mitotic regulation of H3K9ac in which H3K9 deacetylation is a stepwise process - at prophase HDAC2 modulates most transcription-associated H3K9ac-marked loci and at metaphase HDAC3 maintains the reduced acetylation, whereas SIRT1 potentially regulates

H3K9ac by impacting HAT activity. Although the results presented in the MS are of potential interest and this MS provides a characteristic pattern about regulation of H3K9ac from extensive data, there are several technical issues which require to be addressed before accepted for publication in Life Science Alliance.

We thank the reviewer for appreciating the value of our study.

For main point 1 : Identification of the deacetylases involved in modulating H3K9ac during mitosis.

The major issue: The identification of histone deacetylases modulating H3K9ac during mitosis using only specific inhibitory treatments is insufficient, and it is necessary to construct knockdown cell lines for the screened HDAC2, HDAC3 and Sirt1 to demonstrate that mitotic H3K9ac is controlled by phase-specific activity of HDAC2, HDAC3 and SIRT1. The timeframe required is about four months.

While we agree with the reviewer that such knockdown experiments would provide supporting evidence to our observations, there are several key challenges with such an approach. In particular, we are interested in inhibition of the HDAC only during mitosis. Knockdown cell lines would not limit the perturbation to mitosis and would reduce the activity of each deacetylase throughout the cell cycle. Further, in our hands, generation of knockdown constructs and cells lines would require at least 6 months, and thus is beyond the scope of this study.

For main point 2 : Variations in the mitotic chromatin association of key histone deacetylases and histone acetyltransferases.

The major issue: Identification of the chromatin localization patterns of key HATs and HDACs during mitosis. Further immunofluorescence co-localization analysis of HDAC3, HAT1 and EP300 signals detected in metaphase is required to explain variations in the mitotic chromatin association of key histone deacetylases and histone acetyltransferases. The timeframe required is one month.

We thank the reviewer for the interesting suggestion. While we agree that it would be interesting to identify co-localization of these HATs and HDACs. We performed colocalization IF experiments with HDAC3 and p300, which shows that in spite of their retention on the mitotic chromosomes, they have a different binding pattern – HDAC3 is localized in a condensed manner, while EP300 association appears diffused (**Fig. 5**). We have added a sentence in line 219 indicating this point.

Fig. 5. Colocalization of HDAC3 and EP300 on mitotic chromosomes. IF of both EP300 and HDAC3 revealed partial colocalization of both proteins on the mitotic chromosomes. This is true both for metaphase (marked by arrows) and prophase (marked by arrowheads) chromosomes. In both cases EP300 is shows a diffused binding pattern while HDAC3 appears condensed.

For main point 3 : SIRT1 potentially impacts H3K9ac levels indirectly by modulating HAT activity.

The major issue: p300 and HAT-1 were the acetylase, which were both detected on the metaphase chromatin. Experiments about these enzymes with the same treatment as Fig5C are needed to further confirm that SIRT1 indirectly affects H3K9ac levels by potentially modulating specific HAT activity during mitosis. The timeframe required is two months.

We agree with the reviewer that it would be intriguing to identify the HATs that are impacted by SIRT1. Yet, as noted in our response to Reviewer #2, we do not know which of the specific HAT/s is/are the target/s of SIRT1. And although we observed that HAT1 and P300 are retained on the mitotic chromatin, there are multiple HATs that we did not study. Thus, such experiment encompasses too large of a survey to identify the target HAT/s.

For main point 4 : HDAC2 and HDAC3 activity regulates both common and unique genomic loci. The quality of ChIP-seq data provided is acceptable.

We appreciate the reviewer's comment – the ChIP-seq experiments required multiple rounds of optimization to allow quantitative comparison between conditions.

Other issues:

a: The scale of the picture is not marked.

We apologize for this omission and corrected it in the revised manuscript.

b: When measured H3K9ac levels in cells captured at different stages of mitosis, the cell sample size is only 20 cells and should be at least more than 100 cells in order to reduce the error of quantification of the immunofluorescence results.

We agree with the reviewer and apologize for not clarifying the limitations in the original manuscript, yet, it should be noted that in spite the relatively low number of observations, the differences are statistically significant.

The analysis requires cells in different mitotic stages and obtaining 100 cells for each stage entails high volume of image screening. Thus, we repeated the experiments focusing only on metaphase cells and collected 88, 128, 91 and 103 cells for NT, CAY10683, MS-275 and RGFP966 treatments respectively (**Supplementary Figure S2**). As these results were captured using a different microscope setting (the light source in our microscope had to be replaced), we could not combine these data with the previous measurements, and they are presented as a separate analysis.

Thanks to the reviewer's suggestion, we were able to refine our initial conclusion as we observed a small impact (as opposed to the no impact in our original report) of treatment by HDAC2i on H3K9ac signal at metaphase. This is reflected now in the text (**line 164**).

*c: Although it is mentioned in the manuscript that the significant difference analysis is based on $P < 10^{-23}$; one sided t test, the degree of significance should be marked with * in the figure.*

We apologize for this omission and we corrected it in the revised manuscript.

d: There are subscript errors in the drug name and chemical composition in the manuscript, such as line 417, 421, which need to be corrected.

We apologize for these errors, and appreciate the reviewer's note. We corrected these errors in the revised manuscript.

f: Exposure parameters for immunofluorescence microscopy need to be provided to avoid false significance due to overexposure.

We added in the method section the following phrase – “immunofluorescent signals were viewed using Olympus fluorescence microscope using CellSens software. All images were taken under fixed scaling and were normalized to only secondary background control to avoid false significance due to overexposure” (**lines 509-511**).

g: High-resolution, undistorted images are required, especially the cytology images in Figure 2.

We apologize for this corrected it in the revised manuscript.

July 26, 2022

RE: Life Science Alliance Manuscript #LSA-2022-01433-TR

Prof. Itamar Simon
Hebrew University of Jerusalem
Microbiology and Molecular Genetics
Hadassah Ein Kerem
Jerusalem, Please select: 91101
Israel

Dear Dr. Simon,

Thank you for submitting your revised manuscript entitled "Mitotic H3K9ac is controlled by phase-specific activity of HDAC2, HDAC3 and SIRT1". We would be happy to publish your paper in Life Science Alliance pending final revisions necessary to meet our formatting guidelines.

- Please indicate the technical limitations of the study in the discussion session as requested by Reviewer 1
- please address Reviewer 2' point #2 and #3 through discussion as requested
- please upload both your main and supplementary figures as single files
- please add the Twitter handle of your host institute/organization as well as your own or/and one of the authors in our system
- please add the ORCID ID for the corresponding author-you should have received instructions on how to do so
- please add a section for your figure legends, including the main, supplementary, and table legends, to your main manuscript text
- please add a figure callout for Figure S3, Figure S5, and Figure S6 to the main manuscript text

Figure Check:

- -Figure S7 needs scale bars
- -there is a splice in Figure 3D, 1st set of blots, 3rd row (HDAC-3) and 4th row (H3); might also be a splice in top row of 2nd set of blots. Please provide source data for this figure

A. FINAL FILES:

B. MANUSCRIPT ORGANIZATION AND FORMATTING:

Sincerely,

Reviewer #1 (Comments to the Authors (Required)):

The authors addressed most of my concerns and improved the discussion and interpretation of their data in light of current literature. I always find it a good idea to clearly state the technical limitations of the study in the discussion session. Other than that, I have not further comments and suggestions.

Reviewer #2 (Comments to the Authors (Required)):

Overall, I think the authors have disregarded far too many comments from all referees. Below I will limit myself to the comments I made initially:

1/ The inclusion of DSG as a fixative is a good addition. It is however unfortunate the authors did not try to use GFP fusion proteins and live imaging, which is really the gold-standard to address this question. I disagree with the notion that such experiments are "beyond the scope of this study". Moreover, claiming that such assays would take 4-6 months appears to me extremely exaggerated, particularly when transient ectopic transfections would have been sufficient. This is, I believe, a strong weakness of this paper.

2/ It is not clear to me why the authors have not checked the effect of Sirt1 in asynchronous cells; after all, the way the experiments are performed does not fully rule out that the mitotic observations are mere inheritances of effects established during the previous interphase. Also, the authors should have more directly addressed by concern regarding Sirt1 and Hdac3 relationships, if not experimentally at least discussing it thoroughly in the ms main text.

3/ I am not convinced by the reply to my request of discussing more thoroughly mechanistic options. I note also that none of the concepts I requested to discuss (HAT1-cytoplasmic H4 and how acetylation operates if H3K9 acetylases are indeed evicted from the chromosomes) were included in the text, which I find detrimental for the ms.

Reviewer #3 (Comments to the Authors (Required)):

The authors have well addressed all the questions and I feel that this work is now appropriate for publication.

August 1, 2022

RE: Life Science Alliance Manuscript #LSA-2022-01433-TRR

Prof. Itamar Simon
Hebrew University of Jerusalem
Hadassha Ein Kerem
Jerusalem, Please select: 91101
Israel

Dear Dr. Simon,

Thank you for submitting your Research Article entitled "Mitotic H3K9ac is controlled by phase-specific activity of HDAC2, HDAC3 and SIRT1". It is a pleasure to let you know that your manuscript is now accepted for publication in Life Science Alliance. Congratulations on this interesting work.

DISTRIBUTION OF MATERIALS:

Again, congratulations on a very nice paper. I hope you found the review process to be constructive and are pleased with how the manuscript was handled editorially. We look forward to future exciting submissions from your lab.

Sincerely,
